# The Relationship between Neutrophil–Lymphocyte Ratios with Nutritional Status, Risk of Nutritional Indices, Prognostic Nutritional Indices and Morbidity in Patients with Ischemic Stroke

**DOI:** 10.3390/nu16081225

**Published:** 2024-04-20

**Authors:** Naile Fevziye Misirlioglu, Nedim Uzun, Gulenay Defne Ozen, Mustafa Çalik, Ertugrul Altinbilek, Necmettin Sutasir, Sena Baykara Sayili, Hafize Uzun

**Affiliations:** 1Department of Biochemistry, Gaziosmanpaşa Training and Research Hospital, University of Health Sciences, Istanbul 34098, Turkey; n.misirlioglu@saglik.gov.tr; 2Department of Emergency, Gaziosmanpaşa Training and Research Hospital, University of Health Sciences, Istanbul 34098, Turkey; nedim.uzun@sbu.edu.tr (N.U.); mustafa.calik@sbu.edu.tr (M.Ç.); 3Department of Psychology, McGill University, Montreal, QC H3A 1G1, Canada; daphne.oz@mail.mcgill.ca; 4Department of Emergency, Sisli Hamidiye Etfal Education and Research Hospital, University of Health Sciences, Istanbul 34098, Turkey; ertugrul.altinbilek@sbu.edu.tr (E.A.); m.sutasir@saglik.gov.tr (N.S.); 5Emergency Department, Istanbul Training and Research Hospital, Istanbul 34075, Turkey; sena.baykarasayili@saglik.gov.tr; 6Department of Medical Biochemistry, Faculty of Medicine, Istanbul Atlas University, Istanbul 34403, Turkey

**Keywords:** ischemic stroke, prognostic nutritional index, nutritional risk index, controlling nutritional status, systemic immune inflammation index, neutrophil–lymphocyte ratio

## Abstract

**Background:** In recent years, whole blood parameters and derivatives have been used as prognostic criteria in the course of various diseases. The aim of this study was to evaluate the relationship between parameters such as the neutrophil–lymphocyte ratio (NLR), the systemic immune-inflammation index (SII), the prognostic nutritional index (PNI), controlling nutritional status (CONUT) score, nutritional risk index (NRI) and immunonutrition status and disease activity in patients with ischemic stroke of the small-vessel, large-vessel and other etiologies. **Methods:** We retrospectively evaluated the records of 1454 consecutive ischemic stroke patients hospitalized in the emergency department of Gaziosmanpasa Education and Research Hospital from 2019 to 2023. **Results:** Of the 1350 patients with ischemic stroke included in the study, 58.8% had small-vessel disease, 29.3% had large-vessel disease and 11.9% had other etiologies. There was a significant difference between the three etiology groups for PNI and CONUT. The mean of PNI was 47.30 ± 8.06 in the other etiology group, 37.25 ± 7.23 in the small-vessel group, and 34.78 ± 8.16 in the large-vessel disease group. The mean of CONUT was 5.49 ± 1.20 in the small-vessel group, 5.12 ± 1.46 in the large-vessel group and 4.22 ± 1.11 in the other etiology group. In addition, CONUT and PNI were also found to be independent risk factors for mortality. A negative significant correlation was observed between PNI and NLR (r: −0.692), SII (r: −0.591), and CONUT (r: −0.511). Significant correlations were observed between CONUT and NLR (r: 0.402), SII (r: 0.312). **Conclusions:** PNI, CONUT and NRI were found as more accurate prognostic indicators of nutritional status in patients with ischemic stroke. NLR and SII may be important predictive markers in the course and prognosis of stroke.

## 1. Introduction

A cerebrovascular accident (CVA), also known as a stroke, is a focal or generalized neurological deficit that develops suddenly, lasts for more than 24 h, results in death and cannot be explained by any cause other than vascular cause [1,2]. According to the World Health Organization (WHO) definition, a stroke is a focal or global cerebral dysfunction that is of rapid onset, lasts 24 h or longer, or causes death, and has solely vascular causes [3]. When examined in terms of stroke etiology, two types of stroke, ischemic and hemorrhagic, are observed. Of all strokes, 87% are ischemic, 10% are intracerebral hemorrhage and 3% are subarachnoid hemorrhage [4].

Physical, social, and cognitive impairment in patients after stroke can pose a serious problem for quality of life. Malnutrition among stroke patients is an important problem contributing to poor prognosis by causing malnutrition [5]. There are specific objective assessment indices that reflect the immune nutritional status of patients, such as the prognostic nutritional index (PNI) [6], the controlling nutritional status (CONUT) score [7] and the nutritional risk index (NRI).

Since mortality, morbidity and disability rates related to acute ischemic stroke are high, it is important to diagnose these patients early and start treatment [8]. In recent years, much evidence has been presented that shows inflammation after ischemic stroke facilitates acute brain damage. This inflammatory response occurs when the activation of leukocytes and microglia triggers an immune response from peripheral blood to the brain after cerebral ischemia [9]. Complete blood count (CBC), which is an easily accessible, cheap and rapid biochemical test, can be used to evaluate the inflammatory response. Studies have shown that white blood cells (WBC) play a role in the pathogenesis of atherothrombotic stroke as an inflammatory response [8,9,10].

In recent years, an index reflecting both neutrophil elevations reflecting the acute state of inflammation and lymphopenia following acute physiological stress has been used. This index, which is obtained by the neutrophil–lymphocyte ratio (NLR), has been used together with other inflammatory markers in studies and found to be a good indicator of inflammatory status [11]. It has been suggested that NLR may be used as a marker of inflammation in the intensive care unit (ICU) [12]. It has been previously shown that an increase in neutrophil levels in acute coronary syndrome is associated with the extent of myocardial damage and short-term prognosis. In studies, it has been found that early mortality is commonly associated with high NLR values in patients with ischaemic stroke [13,14,15]. In general, PNI is an indicator of immunonutrition and NLR reflects inflammatory status; both have low cost, high accuracy, and high reproducibility with wide application in blood research.

The aim of this study was to evaluate the relationship between parameters such as NLR, systemic immune-inflammation index (SII), CONUT, NRI and PNI with immunonutrition status and disease activity in patients with ischemic stroke of small-vessel, large-vessel, and other etiologies.

## 2. Material and Methods

### 2.1. Study Design and Population

The University of Health Sciences, Gaziosmanpasa Education and Research Hospital, Clinical Research Ethics Committee (Date: 30 March 2022, Code: 49) approval was obtained before the study. The study was conducted in accordance with the criteria of the Declaration of Helsinki, with voluntary participation. Informed consent was obtained from participant or their relative.

The currently widely used “Trial of org. 10,172 in Acute Stroke Treatment” (TOAST) classification includes etiology as well as clinical findings: (i) large artery atherosclerosis, (ii) cardioembolism, (iii) small-vessel disease (lacunar infarcts), (iv) other rare causes of unknown etiology (v) stroke of unknown etiology (cryptogenic), with the last two categories combined for the present analysis [16]. Strokes of unexplained etiology (with no landmark diagnostic finding, in the presence of competing causes or with inadequate diagnostic testing) were referred to as cryptogenic strokes. The 1454 stroke patients included in the study were categorized into 3 groups according to large-vessel disease, small-vessel disease, and other etiologies [16].

In the current study, we classified ischemia as follows:

**(i) Large vessel disease:** Large vessels include both extracranial (main and internal carotid, vertebral) and intracranial arterial system (Willis polygon and proximal branches). Large vessel disease also cardioembolism, Large vessel disease includes large artery atherosclerosis with cardioembolism. **(ii) Small vessel disease:** This affects the intracerebral arterial system, especially the penetrating arteries arising from the distal vertebral artery, basilar artery, middle cerebral artery and arteries of the circle of Willis. **(iii) Other etiologies:** This group includes primary and secondary vasculitis of the central nervous system, rare small-vessel diseases such as CADASIL (cerebral autosomal dominant arteriopathy subcortical infarcts and leukoencephalopathy) and cerebral amyloid angiopathy, congenital vascular diseases, mitochondrial diseases, trauma and dissection and blood diseases. Causes of unidentified etiology: This group includes cerebral infarcts whose etiology cannot be found despite detailed investigations and cases that cannot be adequately investigated. In addition, cases with more than one etiologic cause were evaluated in this group.

Patients were admitted to the emergency department of Gaziosnpasa Education and Research Hospital between 1 January 2019 and 1 June 2023. Chronic diseases of the patients were listed as hypertension (HT), diabetes mellitus (DM), atrial fibrillation, coronary artery disease (CAD), and hyperlipidemia.

Electronic records of patients hospitalized for ischemic stroke were scanned from the hospital’s electronic database system. Neurologic evaluations of the patients were assessed with the National Institutes of Health Stroke Scale (NIHSS) [17].

### 2.2. Inclusion Criteria

Patients with acute ischemic stroke confirmed by cerebral computed tomography (CT) or magnetic resonance imaging and peripheral blood sampling within 24 h of stroke onset and 20 years and older were included.

### 2.3. Exclusion Criteria

Patients with arrest on presentation to the emergency department.Patients admitted after >24 h.Patients with hemorrhagic stroke.Patients with trauma-induced stroke.Patients with a history of intracranial mass.Patients with known hematologic disorders.Patients with a known history of thyroid disease.Patients with chronic renal failure, liver failure.Patients with concurrent acute coronary syndrome (ACS), pulmonary embolism (PE), or acute renal failure (AKF).Patients for whom adequate laboratory data are not available.

### 2.4. Laboratory Parameters

Blood samples were collected within 24 h of stroke onset in standard tubes containing no anticoagulant for CBC parameters and dipotassium ethylenedinitrotetraacetic acid (EDTA) for biochemical parameters. The result of CBC was recorded with an automatic hematology analyzer (Sysmex, Sysmex XN-1000, Norderstedt, Germany). The SII was calculated as (platelet count × neutrophil count)/lymphocyte count [18]. NLR, LMR, and PLR were calculated from neutrophil/lymphocyte/monocyte/thrombocyte count. Routine biochemical parameters such as glucose, cholesterol, and albumin in blood were measured with an automated analyzer (COBAS 8000, ROCHE-2007, Tokyo, Japan). Serum CRP levels were measured nephelometrically (Siemens-Dimention, Munich, Germany). During the study period, the system was checked daily with 2 different levels of internal quality control samples and also with an external quality control program.

### 2.5. Nutritional Indices

Ideal body weight was calculated using the Lorentz formula: height (cm) − 100 − [(height (cm) − 150)/4)] for men and height (cm) − 100 − [(height (cm) − 150)/2.5)] for women. Body mass index (BMI) was calculated and classified according to the World Health Organisation (WHO) [19].

PNI is a scale used to evaluate the nutritional risk of patients, developed using the patient’s blood albumin and lymphocyte count values. The index score is calculated with the formula: serum albumin (g/dL) + (5 × total lymphocyte count (10^9^/L)). A low PNI score (<46.8) indicates high nutritional risk, while a high PNI score (>46.8) indicates low nutritional risk [20].

CONUT score = Serum albumin score + Total lymphocyte count score + Total cholesterol score. In CONUT scoring, total lymphocyte count, serum albumin amount and serum total cholesterol count were used to score between 0 and 12 points. A score of 1 was recorded as normally fed, 2–4 as mild, 5–8 as moderate and 9 and above as severe malnutrition (Table 1) [21].

The NRI was developed to assess the nutritional status of patients in a practical way using objective parameters and is calculated based on body weight and serum albumin level [22]. NRI = [(1.519 × serum albumin, g/dL) + (41.7 × weight (kg)/ideal body weight (IBW; kg)] [23]. A lower NRI indicates a higher risk of malnutrition [19].

### 2.6. Statistical Analysis

Statistical Package for the Social Sciences version 21.0 software package for Windows (IBM Corp., Armonk, NY, USA) and Jamovi 2.4.11 were used for data evaluation and analysis. Frequencies (*n*) and percentages (%) were used to present the descriptive characteristics of the data while numerical variables were represented as median (25th percentile–75th percentile). A chi-square test was used to evaluate the distribution among categorical variables. Whether the data were normally distributed was analyzed through visuals (histograms and Q-Q plots), descriptive techniques (coefficient of variation, skewness, and kurtosis) and analytical methods (Kolmogorov–Smirnov Test). One-way ANOVA or Kruskal–Wallis tests were used for a comparison of continuous variables between more than two groups; adjusted *p* values and Tukey-HSD were used for post hoc significance. The logistic regression analysis was used to determine the risk factors for stroke-related mortality. Results were represented as odds ratio (OR) and 95% confidence interval (95% CI). For multicollinearity, CONUT and PNI, which are highly correlated variables, were not included in the same model. The Pearson or Spearman correlation analyses were used to evaluate the relationship between the numerical variables. A *p* value < 0.05 was considered for statistical significance.

## 3. Results

Of the 1350 patients with ischemic stroke included in the study, 58.8% had small-vessel, 29.3% had large-vessel and 11.9% had other etiologies (Figure 1). Clinical and laboratory characteristics of the patients according to the etiology groups are shown in Table 2 and Table 3.

The proportion of males was 52% in the small-vessel etiology group, 39.1% in the large-vessel group and 88.8% in the other etiology group and there was a significant difference between them (*p* < 0.001). While the frequency of hypertension was similar among the three etiology groups (*p*: 0.112), the frequency of diabetes and dyslipidemia was higher in the small-vessel group than in the large-vessel group (*p*: 0.027; *p*: 0.021, respectively). The frequency of smoking and alcoholism was significantly higher in the other etiology group compared to the small-vessel and large-vessel groups (*p*: <0.001; <0.001, respectively). Mortality rate was 31.9% in patients with ischemic stroke, 19.8% in small-vessel etiology, 20% in other etiologies, and 61.1% in large-vessel etiologies. Mortality was statistically significantly higher in the large-vessel etiology group (*p* < 0.001).

NLR was significantly lower in the other etiology group compared to the small-vessel and large-vessel groups (1.27 (0.75–1.83); 4.46 (3.81–5.18); 4.63 (3.15–5.69); *p* < 0.001). SII was significantly lower in the other etiology group compared to the small-vessel and large-vessel groups (243.34 (142.97–367.66); 871.04 (650.62–1102.69); 898.17 (565.27–1165.79); *p* < 0.001). There was a significant difference between the three etiology groups for PNI (*p* < 0.001). The mean of PNI was 47.30 ± 8.06 in the other etiology group, 37.25 ± 7.23 in the small-vessel group, and 34.78 ± 8.16 in the large-vessel group. The mean of NRI was 62.33 62.02 ± 7.70 in the large-vessel group, 55.33 ± 9.06 in the small-vessel group and 54.78 ± 11.49 in the other etiology group (*p* < 0.001). There was a significant difference between the three etiology groups for CRP (*p* < 0.001). CRP median was 108.83 (66.42–247.63) in the large-vessel group, 46.5 (18.97–82.84) in the small-vessel group and 69.2 (42.09–99.83) in the other etiology group. CONUT scores were also different between the three etiology groups (*p* < 0.001). The mean of CONUT was 5.49 ± 1.20 in the small-vessel group, 5.12 ± 1.46 in the large-vessel group and 4.22 ± 1.11 in the other etiology group (Table 3).

Table 4 shows the results of univariate and multivariate logistic regression analysis for to evaluate risk factors for mortality. Due to the multicollinearity problem, two different logistic regression models including CONUT or PNI were applied. CONUT and PNI were identified as significant predictors for mortality. In addition, etiology, diabetes, dyslipidemia, age, diastolic blood pressure were significant parameters. NLR was also found to be a significant parameter for mortality in the model with CONUT.

Table 5 shows the association between NLR, SII, PNI, NRI and CONUT in the whole group and in the aetiology subgroups. A very weak negative correlation was observed between NRI and PNI only in the whole group (r: −0.100; *p* < 0.001) and in the other etiology group (r: −0.167; *p*: 0.035). In the whole group, a negative correlation was observed between PNI and NLR (r: −0.692; *p* < 0.001), SII (r: −0.591; *p* < 0.001), CONUT (r: −0.511; *p* < 0.001); in addition, significant correlation was observed between these variables in three etiology subgroups. Significant correlations were observed between CONUT and NLR (r: 0.402; *p* < 0.001), SII (r: 0.312; *p* < 0.001) in the whole group; these correlations were also observed in the large-vessel and small-vessel groups.

## 4. Discussion

The main findings of the present study are as follows: (i) there was a significant difference between the three etiology groups for PNI and CONUT; (ii) CONUT and PNI were also found to be independent risk factors for mortality; (iii) a negative significant correlation was observed between PNI and NLR, SII, CONUT; (iv) significant correlations were observed between CONUT and NLR with SII. Nutrition and inflammation play a role in the development of cerebrovascular diseases. Malnutrition in patients with cerebrovascular events is caused by underlying comorbidities, medications and inflammation. Thus, it could be used as a reliable nutritional and inflammation risk stratification tool in patients with cerebrovascular diseases.

In the current study, the frequency of smoking and alcoholism was significantly higher in the other etiology group compared to the small-vessel and large-vessel groups. It suggests a potential association between these lifestyle factors with small-vessel and large-vessel groups. These factors could contribute to or be associated with the development of strokes in this specific subgroup [24].

The survival rates are associated with different etiologies of strokes [24]. In the current study, a breakdown of the mortality rates for each etiological group: large-vessel etiology: 61.1%; small-vessel etiology: 19.8%; other etiologies: 20%; large-vessel etiologies: 38.9%. These rates suggest variations in survival outcomes depending on the underlying cause of the stroke. It is important to note that survival rates can be influenced by various factors, including the severity of the stroke, promptness of medical intervention, the effectiveness of treatments, and individual patient characteristics. A lower survival rate in the large-vessel etiologies group may indicate that strokes with large-vessel involvement tend to be more severe or have a higher risk of complications. Conversely, higher survival rates in other groups may suggest better prognoses associated with those specific etiologies [25].

In the current study, NLR and SII was significantly lower in the other etiology group compared to the small-vessel and large-vessel groups. In the context of stroke, elevated NLR and SII has been linked to worse outcomes, including increased risk of stroke severity, larger infarct size, and higher mortality rates. Maestrini et al. [26] reported that circulating neutrophil counts and NLR were correlated with stroke severity and stroke outcome in patients with acute cerebral ischemia. Higher neutrophil counts and NLR were also independently associated with poorer outcomes and higher mortality rate at 3 months [26]. Researchers are still investigating the precise role of NLR in stroke pathophysiology and its potential as a prognostic marker and therapeutic target. However, its simplicity and availability through routine blood tests make it an attractive candidate for further study in stroke research and clinical practice. In a study by Agard et al. [27], the researchers reported that NLR and SII are easily obtained biomarkers that can be used in early clinical decision making in cases of mild acute ischemic stroke with negative CT scan upon admission. Modulation of the inflammatory cell function plays a role in repairing brain damage after ischemia. Systemic inflammatory response may be involved in the prognosis of acute ischemic stroke [28,29]. Researchers are still investigating the precise role of NLR and SII in stroke pathophysiology and its potential as a prognostic marker and therapeutic target. However, its simplicity and availability through routine blood tests make it an attractive candidate for further study in stroke research and clinical practice.

Malnutrition is a serious public health problem that is associated with adverse outcomes in in ischemic stroke patients. In the current study, there was a significant difference between the three etiology groups for PNI and the NRI score was highest in the large-vessel group. In stroke patients, malnutrition and poor nutritional status can exacerbate complications, impair recovery, and increase the risk of mortality. Therefore, the NRI and PNI serve as a valuable tool for healthcare professionals to identify patients who may benefit from nutritional interventions and support [30,31,32]. Studies have shown that lower NRI scores are associated with worse outcomes in stroke patients, including increased mortality, longer hospital stays, and higher rates of complications. By assessing nutritional risk using the NRI, healthcare providers can implement appropriate interventions, such as dietary modifications, nutritional supplementation, and interdisciplinary care, to optimize outcomes and improve the quality of life for stroke patients. Overall, the PNI and NRI are useful tools in the management of stroke patients, helping healthcare providers identify and address nutritional risk factors that may impact recovery and outcomes.

In the current study, there was a significant difference between the three etiology groups for the CONUT score, which was highest in the small-vessel group. By using the CONUT score to assess nutritional status in stroke patients, healthcare providers can identify those who may benefit from nutritional interventions such as dietary modifications, oral nutritional supplements, or enteral feeding. Early identification and management of malnutrition can help improve outcomes and quality of life for stroke patients. Overall, the CONUT score serves as a valuable tool in the comprehensive care of stroke patients, helping healthcare providers address nutritional needs and optimize recovery [33,34,35].

When evaluating the risk factors for mortality related to ischemic stroke with a large-vessel etiology, several factors such as severity of occlusion, extent of ınfarction, reperfusion status, presence of comorbidities, and age may play significant roles. In the current study, diabetes, dyslipidaemia, age, PNI, and CONUT were also found as risk factors. In addition, BMI and diastolic blood pressure were found to be protective factors. Understanding these risk factors can help healthcare providers identify patients at higher risk of mortality following ischemic stroke with a large-vessel etiology and implement appropriate interventions to optimize outcomes and reduce mortality rates. Early recognition, aggressive management, and comprehensive post-stroke care are essential for improving survival in this high-risk population.

The factors determining prognosis in this disease group are as follows: patient-related parameters, variables monitored in the clinical follow-up of the patient, parameters related to the diagnosis and treatment methods of the disease. Among the patient-related variables, many factors such as age, gender, ethnic characteristics, education level, family history, risk factors, support from family and social environment, genetic factors and biomarkers have been identified. In the current study, due to the multicollinearity problem, CONUT and PNI were identified as important predictors for mortality according to two different logistic regression models including CONUT or PNI. In addition, etiology, diabetes, dyslipidemia, age, diastolic blood pressure were important parameters. NLR, PNI, SII, CONUT were the biomarkers most consistently associated with poor outcomes and demonstrated an additional prognostic value to established prognostic factors and may therefore represent promising targets to be investigated in future prospective studies. Predicting stroke prognosis cannot include absolute determinations due to the unique conditions of each patient, the heterogeneity of the clinic and the difficulty in mastering the variables. Studies with a large number of patients in which more variables are considered together will be guiding in this regard.

## 5. Conclusions

Our study suggested the role of PNI, CONUT and NRI as a more accurate prognostic indicator of nutritional status in patients with ischemic stroke of small-vessel, large-vessel and other etiologies. NLR and SII may be important predictive markers in the course and prognosis of stroke. For each marker, comparing mean values or distributions between the two stroke groups can provide insights into potential differences in nutritional and inflammatory status associated with different stroke etiologies. Additionally, correlation analysis can be performed to assess the strength and direction of the relationship between each marker and stroke etiology within each group separately. These analyses can help identify markers that may serve as predictors of stroke severity or prognosis and contribute to our understanding of the underlying mechanisms of large-vessel and small-vessel strokes. Nutritional therapy in ischemic patients should be a key component of stroke treatment. However, these data need to be verified with further studies to be carried out in the literature.

## Figures and Tables

**Figure 1 nutrients-16-01225-f001:**
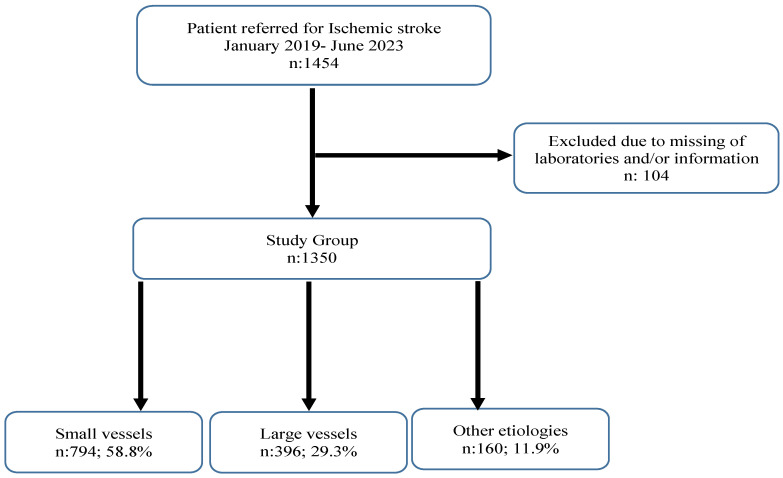
A flow chart of the selection of cases.

**Table 1 nutrients-16-01225-t001:** Assessment of nutritional status with CONUT.

CONUT Score	Level of Malnutrition			
	Normal	Mild	Moderate	Severe
Serum Albumin (g/dL)	≥3.5	3–3.4	2.5–2.9	<2.5
Score	(0)	(2)	(4)	
Total Lymphocytes (×10^9^/L)	≥1600	1200–1599	800–1199	<800
Score	(0)	(1)	(2)	(3)
Total Cholesterol (mg/dL)	≥180	140–179	100–139	<100
Score	(0)	(1)	(2)	(3)
Total Score	(0–1)	(2–4)	(5–8)	(9–12)

**Table 2 nutrients-16-01225-t002:** The distribution of gender and clinical characteristics according to etiology groups.

	All Groups	Small Vessels (*n*: 794; 58.8%)	Large Vessels (*n*: 396; 29.3%)	Other Etiologies (*n*: 160; 11.9%)	
Gender	*n* (%)	*n* (%)	*n* (%)	*n* (%)	*p* Value
Male	710 (52.60%)	413 (52.00%) ^a^	155 (39.10%) ^b^	142 (88.80%) ^c^	<0.001
Female	640 (47.40%)	381 (48.00%)	241 (60.90%)	18 (11.30%)	
Hypertension	1046 (77.50%)	630 (79.30%)	293 (74.00%)	123 (76.90%)	0.112
Diabetes	649 (48.10%)	399 (50.30%) ^a^	168 (42.40%) ^b^	82 (51.30%) ^a,b^	0.027
Dyslipidemia	1101 (81.60%)	667 (84.00%) ^a^	309 (78.00%) ^b^	125 (78.10%) ^a,b^	0.021
Smoking	784 (58.10%)	457 (57.60%) ^a^	201 (50.80%) ^a^	126 (78.80%) ^b^	<0.001
Alcoholism	353 (26.10%)	202 (25.40%) ^a^	82 (20.70%) ^a^	69 (43.10%) ^b^	<0.001
Mortality	431 (31.90%)	157 (19.80%) ^a^	242 (61.10%) ^b^	32 (20.00%) ^a^	<0.001

Chi-square test was applied. Lowercase letters are superscripted ^(a,b,c)^ to identify which variables are differed significantly.

**Table 3 nutrients-16-01225-t003:** The comparison of personal and laboratory characteristics of etiology groups.

	All Groups	Small Vessels	Large Vessels	Other Etiologies	
	Mean ± Std or Median (25p–75p)	Mean ± Std or Median (25p–75p)	Mean ± Std or Median (25p–75p)	Mean ± Std or Median (25p–75p)	*p* Value
Age (years)	64.38 ± 16.43	64.80 ± 16.59	64.30 ± 16.65	62.54 ± 14.96	0.280
Body Mass Index (BMI) (kg/m^2^)	28.30 ± 5.26	27.23 ± 4.97 ^b^	31.01 ± 4.33 ^a^	26.94 ± 6.24 ^b^	<0.001
Temperature (°C)	36.99 ± 0.46	37.0 ± 0.46	36.99 ± 0.44	36.94 ± 0.45	0.397
Systolic Blood Pressure (mmHg)	153 (143–163)	153 (141–157) ^a^	154 (144–164) ^b^	154 (144–163) ^a,b^	0.005
Diastolic Blood Pressure (mmHg)	85 (79–88)	84 (79–88) ^a^	85 (80–89) ^b^	85 (79–88) ^a,b^	0.003
NIHSS score	7 (6–8)	7 (5–8)	7 (6–8)	7 (6–8)	0.160
Lymphocytes (×10^9^/L)	1743.5 (1459–2074)	1693.5 (1459–1973) ^a^	1667 (1348.5–1998) ^a^	4200 (2985–5370) ^b^	<0.001
Neutrophils (µL)	7786 (6343–8212)	7832.5 (7535–8194) ^a^	7871 (6330–8457) ^a^	4670 (3510–6900) ^b^	<0.001
Neutrophil–lymphocyte ratio (NLR)	4.23 (2.96–5.16)	4.46 (3.81–5.18) ^a^	4.63 (3.15–5.69) ^a^	1.27 (0.75–1.83) ^b^	<0.001
Systemic immune-inflammation index (SII)	807.68 (506.32–1092.19)	871.04 (650.62–1102.69) ^a^	898.17 (565.27–1165.79) ^a^	243.34 (142.97–367.66) ^b^	<0.001
White blood cell (10^3^/µL)	9.27 ± 2.03	9.29 ± 2.04	9.15 ± 1.99	9.53 ± 2.07	0.127
Platelet (10^9^/L)	201.55 ± 47.81	201.38 ± 47.31	202.10 ± 49.05	201.05 ± 47.44	0.962
Fasting glucose (mg/dL)	120 (102–145)	120 (102–145)	120 (100–145)	120 (102–145)	0.971
Albumin (g/dL)	2.61 (2.55–2.7)	2.62 (2.59–2.72) ^b^	2.6 (2.29–2.69) ^a^	2.61 (2.54–2.71) ^b^	<0.001
Prognostic nutritional index (PNI)	37.71 ± 8.45	37.25 ± 7.23 ^a^	34.78 ± 8.16 ^b^	47.30 ± 8.06 ^c^	<0.001
Nutritional risk index (NRI)	57.23 ± 9.53	55.33 ± 9.06 ^b^	62.02 ± 7.70 ^a^	54.78 ± 11.49 ^b^	<0.001
Total cholesterol (mg/dL)	159 (146–267)	149 (140–156) ^a^	276 (194–527) ^b^	317.5 (278–395) ^a,b^	<0.001
C-reactive protein (CRP) mg/L	66.42 (33.32–100.84)	46.5 (18.97–82.84) ^a^	108.83 (66.42–247.63) ^b^	69.2 (42.09–99.83) ^c^	<0.001
CONUT	5.23 ± 1.33	5.49 ± 1.20 ^a^	5.12 ± 1.46 ^b^	4.22 ± 1.11 ^c^	<0.001

NIHSS, National Institutes of Health Stroke Scale; CONUT, controlled nutritional status. Kruskal–Wallis tests were used for the comparisons. Adjusted *p* values were used to identify which groups differed signifacantly. Lowercase letters are superscripted ^(a,b,c)^ to identify which variables are differed significantly.

**Table 4 nutrients-16-01225-t004:** The univariate and multivariate regression analysis results for to evaluate risk factors for mortality.

	Univariate		Backward (PNI Excluded)		Backward (CONUT Excluded)	
	OR (95% CI)	*p* Value	OR (95% CI)	*p* Value	OR (95% CI)	*p* Value
Etiology (ref: Other etiology)		<0.001		<0.001		<0.001
Small Vessels	0.986 (0.645–1.508)	0.948	2.355 (1.428–3.885)	0.001	3.664 (2.118–6.339)	<0.001
Large vessels	6.286 (4.061–9.73)	<0.001	24.076 (13.955–41.535)	<0.001	49.503 (26.526–92.383)	<0.001
Diabetes	1.159 (0.922–1.457)	0.207	1.467 (1.120–1.921)	0.005	1.517 (1.146–2.009)	0.004
Dyslipidemia	1.643 (1.196–2.258)	0.002	2.059 (1.414–2.996)	<0.001	1.753 (1.193–2.576)	0.004
Age (years)	1.008 (1.001–1.015)	0.024	1.010 (1.002–1.019)	0.013	1.013 (1.004–1.021)	0.004
Diastolik Blood Preasure (mmHg)	0.989 (0.976–1.002)	0.088	0.980 (0.965–0.995)	0.010	0.982 (0.966–0.998)	0.024
NIHSS score	1.059 (0.985–1.138)	0.121	-	-	-	-
NLR	0.846 (0.793–0.903)	<0.001	0.659 (0.602–0.722)	<0.001	-	-
SII	0.999 (0.999–1.000)	<0.001	-	-	-	-
PNI	1.057 (1.042–1.072)	<0.001	-	-	1.135 (1.110–1.160)	<0.001
CONUT	0.950 (0.872–1.035)	0.244	1.170 (1.051–1.303)	0.004	-	-

**Table 5 nutrients-16-01225-t005:** The correlation analyses between inflammatory markers and nutritional markers in all groups and etiology sub-groups.

	All Group				
**Variables**		**SII**	**PNI**	**NRI**	**CONUT**
**NLR**	r	**0.826 ***	**−0.692 ***	0.034 *****	**0.402 ***
	*p*	**<0.001**	**<0.001**	0.208	**<0.001**
**SII**	r		**−0.591 ***	0.039 *****	**0.312 ***
	*p*		**<0.001**	0.148	**<0.001**
**PNI**	r			**−0.100 ^¶^**	**−0.511 ^¶^**
	*p*			**<0.001**	**<0.001**
**NRI**	r				−0.022 **^¶^**
	*p*				0.418
**Small Vessels**					
		SII	PNI	NRI	CONUT
**NLR**	r	**0.694 ***	**−0.653 ***	−0.036 *****	**0.335 ***
	*p*	**<0.001**	**<0.001**	0.311	**<0.001**
**SII**	r		**−0.455 ***	−0.015 *****	**0.164 ***
	*p*		**<0.001**	0.674	**<0.001**
**PNI**	r			0.023 **^¶^**	**−0.376 ^¶^**
	*p*			0.523	**<0.001**
**NRI**	r				−0.027 **^¶^**
	*p*				0.454
**Large Vessels**					
		SII	PNI	NRI	CONUT
**NLR**	r	**0.837 ***	**−0.577 ***	0.004 *****	**0.389 ***
	*p*	**<0.001**	**<0.001**	0.937	**<0.001**
**SII**	R		**−0.504 ***	0.023 *****	**0.321 ***
	*p*		**<0.001**	0.650	**<0.001**
**PNI**	r			−0.039 **^¶^**	**−0.646 ^¶^**
	*p*			0.444	**<0.001**
**NRI**	r				−0.009 **^¶^**
	*p*				0.853
**Other Etiologies**					
		SII	PNI	NRI	CONUT
**NLR**	R	**0.919 ***	**−0.531 ***	0.107 *****	0.079 *****
	*p*	**<0.001**	**<0.001**	0.178	0.319
**SII**	r		**−0.509 ***	0.090 *****	0.064 *****
	*p*		**<0.001**	0.258	0.424
**PNI**	r			**−0.167 ^¶^**	**−0.446 ^¶^**
	*p*			**0.035**	**<0.001**
**NRI**	r				0.043 **^¶^**
	*p*				0.590

**^¶^**: Pearson correlation analysis was applied. *****: Spearman correlation analysis was applied.

## Data Availability

The data underlying this article are available in the article. If needed, please contact the corresponding author. The email address is huzun59@hotmail.com.

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
