# Peer review of "The Relationship between Neutrophil–Lymphocyte Ratios with Nutritional Status, Risk of Nutritional Indices, Prognostic Nutritional Indices and Morbidity in Patients with Ischemic Stroke"

_nutrients, 2024, doi:10.3390/nu16081225_

Round 1

Reviewer 1 Report

Comments and Suggestions for Authors

The authors have undertaken a detailed analysis of  the potential of whole blood parameters and nutritional status and some of their derivatives to act as prognostic criteria in the course of inflammatory stroke. Their  aim  was  to evaluate the relationship between several parameters and disease activity in patients with various aetiologies of ischemic stroke.  The analysis was undertaken retrospectively and although highly detailed, the statistical analysis is poorly described  and presented and lacks rigour.

1.    Several parameters such as Ferritin, PT, APTT, INR are mentioned in Methods, but I cannot see any results for these-what are their relevance? Equally we are not informed how certain parameters e.g cholesterol or serum albumin were measured.
2.    Table 2 is very confusing and contains several inconsistencies, the least of which is to use p value as  0,157 or 0.157 (there are several examples of this misuse of 0, instead of  the decimal point.
3.    Why present parameters as mean +/-SD and as median and sp? There is rarely justification for this. CRP for instance clearly cannot be presented as mean +/-SD.
4.    Several parameters do not have units associated with them.
5.    Table 2 should follow Figure 1.
6.    The text of the Results is very dense and mainly is a reiterates the Figures and lacks any real expansion of the data that is present in these Figures/Tables.
7.    Regression and correlation results are major outcomes but are minimally described in particular the logistic methods. For instance, the correlation results with several highly significant correlations, suggests that there may be substantial multicollinearity present in the data. Therefore,  was this a case and if so how was it dealt with during regression.
8.    Figure 2, 3 and 4 have highly uninformative legends and are fitting lines to correlation analysis-this is at best pointless and in fact is generally incorrect, lines are obtained from regression analysis not correlation. Figure 2 has a straight line fitted when in fact the line is clearly a curve and may suggest relevant differences in the populations studied and may well inform the regression analysis and must be accounted for.
9.    The Discussion is highly Introductory and includes (for instance) a totally unnecessary description of the role of correlation.

Comments on the Quality of English Language

--

Author Response

Dear Editor,

First, we would like thank the reviewers for the helpful comments, which led us to conduct appropriate experiments. The revised manuscript has subsequently been rewritten to address these concerns and comments of the reviewers.

We are grateful for your understanding and cooperation in this matter.

We believe that the manuscript is now suitable for review. We look forward to your reply.

RESPONSE TO REVIEWERS:

Reviewer 1

Comments and Suggestions for Authors

The authors have undertaken a detailed analysis of  the potential of whole blood parameters and nutritional status and some of their derivatives to act as prognostic criteria in the course of inflammatory stroke. Their  aim  was  to evaluate the relationship between several parameters and disease activity in patients with various aetiologies of ischemic stroke. The analysis was undertaken retrospectively and although highly detailed, the statistical analysis is poorly described  and presented and lacks rigour.

Response: We thank the reviewer for their valuable comments. The analysis has been rigorously reviewed. No erroneous statistics were found, but confusing information in the presentation of some tables has been removed. We have tried to answer and correct the reviewers all comments one by one.

  1. Several parameters such as Ferritin, PT, APTT, INR are mentioned in Methods, but I cannot see any results for these-what are their relevance? Equally we are not informed how certain parameters e.g cholesterol or serum albumin were measured.

Response: Since ferritin, PT, APTT, APTT, INR levels in each patient could not be reached, we did not include them in the tables. It was forgotten to remove them from the method section. Ferritin, PT, APTT, APTT, and INR were excluded from the method section.

Glucose, cholesterol, and albumin were added to the method section.

  1. Table 2 is very confusing and contains several inconsistencies, the least of which is to use p value as 0,157 or 0.157 (there are several examples of this misuse of 0, instead of  the decimal point.

Response: All decimal usage has been corrected. The table can be confusing as we try to give all descriptive data in the table. Unfortunately, when we analyzed our data with both Kolmogorov-Smirnov test and skewness kurtosis values, the data did not meet the normal distribution criteria. Although we tried different transformations in this regard, it still did not meet the normal distribution. We find the reviewer's comment valuable. Therefore, we have decided to remove the mean and standard deviation values to eliminate confusion. If the reviewer has a different suggestion about the table, we are open to implement it.

  1. Why present parameters as mean +/-SD and as median and sp? There is rarely justification for this. CRP for instance clearly cannot be presented as mean +/-SD.

Response: We thank the reviewer for their valuable comments. We gave the median and (25-75th percentile) values in addition to the mean and standard deviation. We used the median (25-75th percentile) for all our interpretations in the results. We included the mean and standard deviation only to indicate that the data is a basic descriptive characteristic and that it is not normally distributed. However, after considering the reviewer's previous comment, we decided to remove the mean and standard deviation.

  1. Several parameters do not have units associated with them.

Units were added.

  1. Table 2 should follow Figure 1.

Response: We thank the reviewer for their valuable comments. We are of the similar opinion. The location of Figure 1 has been corrected.

  1. The text of the Results is very dense and mainly is a reiterates the Figures and lacks any real expansion of the data that is present in these Figures/Tables.

Response: The figures were removed because they did not provide enough information. The findings have also been extensively edited.

  1. Regression and correlation results are major outcomes but are minimally described in particular the logistic methods. For instance, the correlation results with several highly significant correlations, suggests that there may be substantial multicollinearity present in the data. Therefore, was this a case and if so how was it dealt with during regression.

Response: We thank the referee for their comment. To account for multicollinearity, we did so by not putting in the same model those correlated with 0.4 and above in the correlation matrix. This way we had to include CONUT and PNI in a separate model.

Although NLR and SII are also highly correlated with each other, we used them together in the backward model. This is how we dealt with the multicollinearity problem, as the backward model will exclude the insignificant from the model and keep the most important in the model. We revised the logistic regression to take multicollinearity into account. We updated the table. There were no major changes in the results. However, there were minor changes in the ORs.

  1. Figure 2, 3 and 4 have highly uninformative legends and are fitting lines to correlation analysis-this is at best pointless and in fact is generally incorrect, lines are obtained from regression analysis not correlation. Figure 2 has a straight line fitted when in fact the line is clearly a curve and may suggest relevant differences in the populations studied and may well inform the regression analysis and must be accounted for.

Response: We thank the reviewers for their valuable comments. We agree with the reviewers's comment about the figures. That is why we have removed these figures and we have considered adding a multivariate regression analysis instead. However, we encountered an expected multicollinearity between NLR and SII in the regression analyses, and a multicollinearity problem as CONUT and PNI are both nutritional measures. Univariate regression analysis would be a repetition of correlation analyses, so we considered it unnecessary. Unfortunately, we could not perform a multivariate regression for these variables because of the multicollinearity problem. We added this as a limitation to the study.

  1. The Discussion is highly Introductory and includes (for instance) a totally unnecessary description of the role of correlation.

Response: Necessary arrangements have been made.

Reviewer 2 Report

Comments and Suggestions for Authors

This study investigated how prognostic nutritional index (PNI), systemic immune-inflammation index (SII), neutrophil-lymphocyte ratio (NLR), controlling nutritional status (CONUT) score and nutritional risk index (NRI) are related to immunonutrition status and disease activity in patients with ischemic stroke of small vessel, large vessel and other etiologies. 1454 patients with ischemic stroke hospitalized in the emergency department of Gaziosmanpasa Education and Research Hospital, from 2019 to 2023, were retrospectively included in the study. 

The authors concluded that PNI, CONUT and NRI have a role as a prognostic indicator of nutritional status in patients with ischemic stroke, while NLR and SII may be important predictive markers in the course and prognosis of the disease. 

The study, conducted on an appropriate number of people, maybe was done correctly but the methods are not sufficiently explained. For example, it's not clear how you determine stroke etiology. Unfortunately, the influence of nutritional status on stroke prognosis depends, according to you study, to stroke mechanism. So, it's a absolutely mandatory to clarify this point.

Also, it's not useful to introduce your paper with stroke incidence and prevalence. Please erase all epidemiological reference.

Comments on the Quality of English Language

Please, control overall English language. All abbreviation must be explained before the first citation (SVO?). In table 2, please correct systolic and diastolic. Cancel line 231. Table 4, inflammatuar. and so on .....

Author Response

Dear Editor,

First, we would like thank the reviewers for the helpful comments, which led us to conduct appropriate experiments. The revised manuscript has subsequently been rewritten to address these concerns and comments of the reviewers.

We are grateful for your understanding and cooperation in this matter.

We believe that the manuscript is now suitable for review. We look forward to your reply.

RESPONSE TO REVIEWERS:

Review Report (Reviewer 2)

Open Review

( ) I would not like to sign my review report

(x) I would like to sign my review report

Quality of English Language

( ) I am not qualified to assess the quality of English in this paper

( ) English very difficult to understand/incomprehensible

( ) Extensive editing of English language required

( ) Moderate editing of English language required

(x) Minor editing of English language required

( ) English language fine. No issues detected

Yes      Can be improved        Must be improved      Not applicable

Does the introduction provide sufficient background and include all relevant references?

( )        (x)       ( )        ( )

Are all the cited references relevant to the research?

(x)       ( )        ( )        ( )

Is the research design appropriate?

( )        ( )        (x)       ( )

Are the methods adequately described?

( )        ( )        (x)       ( )

Are the results clearly presented?

( )        ( )        (x)       ( )

Are the conclusions supported by the results?

( )        ( )        (x)       ( )

Comments and Suggestions for Authors

This study investigated how prognostic nutritional index (PNI), systemic immune-inflammation index (SII), neutrophil-lymphocyte ratio (NLR), controlling nutritional status (CONUT) score and nutritional risk index (NRI) are related to immunonutrition status and disease activity in patients with ischemic stroke of small vessel, large vessel and other etiologies. 1454 patients with ischemic stroke hospitalized in the emergency department of Gaziosmanpasa Education and Research Hospital, from 2019 to 2023, were retrospectively included in the study.

The authors concluded that PNI, CONUT and NRI have a role as a prognostic indicator of nutritional status in patients with ischemic stroke, while NLR and SII may be important predictive markers in the course and prognosis of the disease.

The study, conducted on an appropriate number of people, maybe was done correctly but the methods are not sufficiently explained. For example, it's not clear how you determine stroke etiology. Unfortunately, the influence of nutritional status on stroke prognosis depends, according to you study, to stroke mechanism. So, it's a absolutely mandatory to clarify this point.

 Also, it's not useful to introduce your paper with stroke incidence and prevalence. Please erase all epidemiological reference.

Necessary arrangements have been made.

Comments on the Quality of English Language

Please, control overall English language. All abbreviation must be explained before the first citation (SVO?). In table 2, please correct systolic and diastolic. Cancel line 231. Table 4, inflammatuar. and so on .....

Necessary arrangements have been made.

Round 2

Reviewer 1 Report

Comments and Suggestions for Authors

Table 2 remains an issue. My previous comment related to the fact that the authors should EITHER report a results as mean +/- SD, OR as median and range depending on distribution but not as both. For some parameters mean and SD are appropriate and for others median and range. The authors should ammend the Table accounting for this.

Comments on the Quality of English Language

There are minor qulity of English issues

Author Response

Dear Editor,

First, we would like thank the reviewers for the helpful comments, which led us to conduct appropriate experiments. The revised manuscript has subsequently been rewritten to address these concerns and comments of the reviewers.

We are grateful for your understanding and cooperation in this matter.

We believe that the manuscript is now suitable for review. We look forward to your reply.

RESPONSE TO REVIEWERS:

Reviewer 1

Comments and Suggestions for Authors

Table 2 remains an issue. My previous comment related to the fact that the authors should EITHER report a results as mean +/- SD, OR as median and range depending on distribution but not as both. For some parameters mean and SD are appropriate and for others median and range. The authors should ammend the Table accounting for this.

Response: We thank the reviewer for their valuable comments. All variables in Table 2 gave a value of p<0.001 in the Kolmogorov-Smirnov test. In accordance with the referee's recommendation, we evaluated the data again with the histogram, Q-Q plots, coefficient of variation, skewness, and kurtosis values. Then, we gave mean± std for age, BMI, temperature, WBC, Platelet, PNI, NRI, CONUT. We also made all corrections in the text.

Reviewer 2 Report

Comments and Suggestions for Authors

Unfortunately, there are still important modifications to be made.

Erase all epidemiologic considerations, please.

It's not necessary to explain NIHS scoring. Viceversa, the CONUT score is not fully detailed.

I syill do not understand how you split stroke etiologies. You cite the TOAST classification but you used another classification: small vessel (I suppose you mean Small Vessel Disease) and large vessel (what do you mean? large vessel occlusion? so you include large arteries atherosclerosis along with cardio-emolism?). Please, be clear. It's mandatory for results interpretation.

Could you design graphs for a more pleasant lecture of your results?

Exclusion criteria: thank you for indicate all criteria but the presentation is not correct nor clear. 

In table 1, whait is 'survival'?

In general, time points for examen collection are missing.

Comments on the Quality of English Language

Please, correct the language with english fluent person. 

Author Response

Dear Editor,

First, we would like thank the reviewers for the helpful comments, which led us to conduct appropriate experiments. The revised manuscript has subsequently been rewritten to address these concerns and comments of the reviewers.

We are grateful for your understanding and cooperation in this matter.

We believe that the manuscript is now suitable for review. We look forward to your reply.

RESPONSE TO REVIEWERS:

Reviewer 2

Comments and Suggestions for Authors

Unfortunately, there are still important modifications to be made.

Erase all epidemiologic considerations, please.

Response: All epidemiologic assessments were deleted.

It's not necessary to explain NIHS scoring. Viceversa, the CONUT score is not fully detailed.

Response: Necessary arrangements have been made.

I still do not understand how you split stroke etiologies. You cite the TOAST classification but you used another classification: small vessel (I suppose you mean Small Vessel Disease) and large vessel (what do you mean? large vessel occlusion? so you include large arteries atherosclerosis along with cardio-emolism?). Please, be clear. It's mandatory for results interpretation.

Response: Necessary arrangements have been made.

Could you design graphs for a more pleasant lecture of your results?

Graphical Abstract

Exclusion criteria: thank you for indicate all criteria but the presentation is not correct nor clear.

Response: Necessary arrangements have been made.

In table 1, whait is 'survival'?

Response: Survival' is the opposite of mortality. We have given mortality rates in the table instead of Survival rates.